# Research on the Impact of Economic Policy Uncertainty on Enterprises' Green Innovation—Based on the Perspective of Corporate Investment and Financing Decisions

Wenjun Zhou [1,2], Xiaorong Huang [1], Hao Dai [1], Yuanmeng Xi [1], Zhansheng Wang [1] and Long Chen [1,2,3,*]

1   School of Management, Hebei GEO University, Shijiazhuang 050031, China; zhouwenjun@hgu.edu.cn (W.Z.); alice_121@163.com (X.H.); daihao0329@outlook.com (H.D.); xiyuanmeng2021@163.com (Y.X.); Ketty616WZS@163.com (Z.W.)
2   Research Center of Natural Resources Assets, Hebei GEO University, Shijiazhuang 050031, China
3   Post-Doctoral Scientific Research Workstation of Hebei Bank, Shijiazhuang 050011, China
*   Correspondence: chenlong85@hgu.edu.cn

**Abstract:** Improving enterprises' green innovation ability is beneficial to realize the "win–win" of economic development and environmental protection. As the global economic situation is complex and volatile, economic policies changed frequently. Will the rising uncertainty of economic policies affect enterprises' green innovation? Taking China's A-share-listed companies from 2008 to 2019 as the research sample, the Baker index based on news media and network information is used to measure the uncertainty of national economic policy, and the official exchange index based on the complex network is used to measure the uncertainty of economic policy in prefecture-level cities. It is found that there is an inverted U-shaped relationship between economic policy uncertainty and firms' green innovation capability. Moreover, the uncertainty index of national macroeconomic policy is mostly on the left side of the inverted U shape, which can promote the improvement of enterprises' green innovation ability. However, too frequent changes in regional economic policies will inhibit enterprises' green innovation ability. This paper further analyzes the moderating effect of financialization of investment behavior and financing constraint on the impact of economic policy uncertainty on green innovation of enterprises from the perspective of investment and financing behavior choice. It is found that the impact of economic policy uncertainty on green innovation is more obvious for firms with low financing constraints and low financialization.

**Keywords:** economic policy uncertainty; green innovation; investment and financing behavior of enterprises

## 1. Introduction

China's economy has achieved world-renowned achievements in the past 40 years, but the long-term extensive economic development model has increased the burden of the ecological environment. Breaking the situation of "one or another" between the economy and the environment is an important part of building an economical and ecological civilization. In order to achieve a win–win situation in economic development and environmental protection, it is necessary to improve the green innovation ability of enterprises. Unlike direct participation in environmental governance and environmental protection investment, green innovation can not only reduce environmental pollution by enterprises and improve environmental performance, but more importantly, green innovation is the key for enterprises to produce green differentiated products, stimulate new market demand, and effectively improve their green competitiveness.

However, solely depending on the market makes it difficult to effectively promote the improvement of enterprises' green innovation capabilities and solve the dilemma of environmental quality improvement and high-quality economic development. In the

practice of market economy, environmental protection and economic development are often manifested in contradictions. Economic growth cannot effectively solve the problem of environmental degradation. Moreover, due to the negative externalities of environmental problems caused by the nature of public goods of environmental resources, the existence of opportunism of microeconomic subjects, and China's long-term high-energy economic growth model, there is a first-mover advantage in non-green production and its technology research and development in the free market economy. There is not enough motivation for enterprises independently carrying out green production and green technology research and development. Therefore, it is difficult to effectively solve the market mechanism itself to stimulate green innovation of enterprises, and the problem of market failure needs to be solved through government intervention.

In order to advance the improvement of the green innovation ability of enterprises to promote the high-quality development of the economy, governments have formulated a large number of economic regulation policies. However, due to the complex and volatile global economic situation and the slowdown in domestic economic growth, the Chinese government frequently adjusts macroeconomic policies, and economic policy uncertainties are rising. Drawing on the Economic Policy Uncertainty (EPU) Index constructed by Baker [1], it can be found that China's Economic Policy Uncertainty Index was higher in 2001–2003, 2008–2009, 2012–2013, 2016–2018, and 2019–2020 (as shown in Figure 1). The high index from 2001 to 2003 was caused by the reform of state-owned enterprises, a series of policies introduced by the government in response to the financial crisis in Southeast Asia, and the economic recession caused by SARS. When the world financial crisis broke out in 2008–2009, the Chinese government launched the "Four Trillion Plan" to avoid a severe economic recession. After that, in order to cope with the superposition of the three phases of "economic growth speed shift period, structural adjustment pain period and early stimulus policy digestion period", the government implemented a series of policies such as "mass entrepreneurship, mass innovation", "three reductions in exchange rate" and so on. The frequent introduction of various policies has increased the uncertainty of economic policies. The rise of the policy uncertainty index in 2012–2013 was caused by the continuous adjustment of economic policies brought about by the change of government. In the period of 2016–2017, in order to better cope with the downward pressure of the transition economy and the complicated political and economic situation at home and abroad, the government formulated a series of policies on import and export, reducing enterprise costs, and improving manufacturing development, which not only brings benefits to enterprises but also increases the uncertainty of economic policies. In 2019–2020, the COVID-19 outbreak, the disruption of global supply chains, and the stagnation of international trade led to a series of policy responses. These policies have increased the uncertainty of economic policy while alleviating the economic downturn in China.

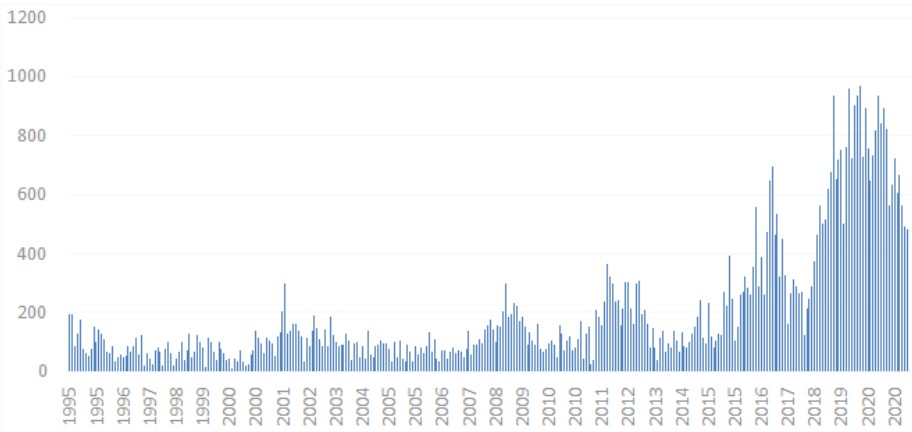

**Figure 1.** China's Economic Policy Uncertainty Index.

Macroeconomic policies will affect the risk preference and investment and financing behavior choice of enterprises. Especially for the green innovation behavior of enterprises, such investment has the characteristics which are large investment amount, long payback period, and large uncertainty of results. Therefore the investment risk is high. The mismatch between profit and risk will lead to the lack of innovation motivation and the decline of innovation ability. Effective macroeconomic policies can reduce the risk of green investment and increase the return of green investment, thus promoting the willingness of enterprises to innovate. However, with increasing economic policy uncertainty, the external environmental risks faced by enterprises increase [2,3]. This will affect enterprises' expectations of future earnings, thus influencing their investment behavior choices. Therefore, the rise of macroeconomic policy uncertainty will have an important impact on enterprises' green innovation behavior.

Does the rising degree of macroeconomic policy uncertainty affect enterprises' green innovation behavior? If so, how? Many scholars at home and abroad have discussed this issue and formed a large number of research results. Some scholars believe that economic policy uncertainty will inhibit the enterprise of green innovation [2,4]. However, some scholars believe that economic policy uncertainty can stimulate enterprises to innovate and improve their innovation ability [5–7]. Therefore, no consensus conclusion has been reached on how macroeconomic policy uncertainty affects enterprises' green innovation behavior. This is also the foothold of this paper and the starting point of research. What is the relationship between economic policy uncertainty and enterprise green innovation? What are the moderating variables that affect the relationship between economic policy uncertainty and firms' green innovation capability? How to determine the frequency of economic policy adjustment to promote the improvement of enterprises' green innovation level? These three issues are discussed and the research framework is constructed to clarify the substantial impact of economic policy uncertainty on green innovation of enterprises.

Different from previous studies which focus more on the impact of macroeconomic policy changes at the national level on enterprise behavior from the perspective of the whole country, this paper constructs economic policy uncertainty indices from the national and regional levels to measure the frequency of economic policy changes. The local government of each province and city has certain policy-making power, and different local regulations have been formulated. Enterprises are not only affected by national economic policies but also must comply with the economic policies set by provincial and municipal governments. Regional economic policy changes will undoubtedly have an impact on the investment behavior of local enterprises. With the changes in economic conditions and industrial structure, the uncertainty of economic policies in different regions also varies. Does the uncertainty of regional economic policy affect enterprises' green innovation? Is there any difference in the impact of regional economic policy uncertainty and national macroeconomic policy uncertainty on the enterprise of green innovation? Which policy changes have a stronger impact on corporate green innovation? The discussion on this issue can enrich the research scope of economic policy uncertainty. Measuring the uncertainty of economic policy from multiple angles will increase the research depth of economic policy uncertainty, which is also the main contribution of this paper.

Based on China's A-share-listed companies from 2008 to 2019 as research samples, the Baker index based on news media and network information is used to measure the uncertainty of national economic policy, and the official exchange index based on the complex network is used to measure the uncertainty of economic policy in prefecture-level cities. It is found that there is an inverted U-shaped relationship between economic policy uncertainty and firms' green innovation capability. The uncertainty of economic policy can not only improve enterprises' green innovation ability by promoting enterprises' R&D investment, but also bring fluctuations to enterprises' external environment, which has a negative impact on enterprises' economic environment and inhibits enterprises' green innovation. Therefore, uncertainty of economic policy has both "incentive effect" and "inhibition effect" on green innovation of enterprises. The results enrich the research on the

relationship between economic policy uncertainty and green innovation. By comparing the influence of national macroeconomic policy uncertainty and regional economic policy uncertainty on firms' green innovation, it is found that national macroeconomic policy uncertainty index is mostly on the left side of the inverted U shape, which can promote firms' green innovation ability. However, too frequent changes in regional economic policies will inhibit enterprises' green innovation ability. Therefore, at present, China's national macroeconomic policy has a more obvious role in promoting the enterprise of green innovation. Local governments should improve the stability of economic policies and reduce frequent changes to economic policies. This paper measures the impact of economic policy uncertainty on the enterprise of green innovation from national and regional levels, making the research on economic policy uncertainty more comprehensive and detailed.

Different from previous studies that focus more on the macro perspective, this paper further analyzes the impact of firm behavior choice on the relationship between economic policy uncertainty and firm green innovation from the micro perspective of firm investment and financing behavior decisions. It is found that the impact of economic policy uncertainty on green innovation is more obvious in low financing constrained enterprises and low financialized enterprises.

The research of this paper can more comprehensively study the impact of economic policy uncertainty on green innovation of enterprises, and provide theoretical basis and policy choice space for the government to clarify the regulation frequency of macroeconomic policies to stimulate enterprises' green innovation vitality and stimulate enterprises' green innovation investment.

## 2. Literature Review

Economic policy uncertainty refers to the fact that economic entities cannot predict with certainty whether, when, and how the government will change economic policies [8]. Since the outbreak of the world financial crisis in 2008, the global economy has repeatedly fluctuated. In order to cope with the complex economic, political, financial, and other aspects of the environment, governments have begun to frequently introduce various economic policies, which has led to economic policy uncertainty, and the policy uncertainty caused by the frequent adjustment of policies has attracted the attention of more and more scholars.

### 2.1. The Theoretical Basis of the Uncertainty of Economic Policy Affecting the Green Innovation of Enterprises

Scholars often study how macroeconomic uncertainty affects macroeconomics and microenterprises based on the three theories of real options, risk compensation, and growth options.

#### 2.1.1. Real Options Theory

The theory of real options is most common in the discussion of uncertainty and economic growth. Bernanke thought that the investment behavior of an enterprise can be regarded as a series of options: when the investment behavior is irreversible, the enterprise needs to weigh the costs and benefits between investing "now" and waiting for better investment opportunities in the future [9]. From the perspective of physical options theory, when the uncertainty of the future rises, it is more valuable for enterprises to choose to postpone investment because, during the waiting period, enterprises may be able to obtain more information about the future to avoid possible large losses [10,11]. Therefore, economic policy uncertainty will significantly reduce investment and output at both the macro and micro levels.

At the macro level, uncertainty will reduce investment and output. Uncertainty shocks had a significant inhibitory effect on GDP growth and investment in the United States [12,13]. Baker et al. constructed an index of economic policy uncertainty and found that economic policy uncertainty would reduce output [1].

For microenterprises, the uncertainty of economic policies means that the external environment risks faced by enterprises are increasing [2,3] and bank credit risks are increased [14], resulting in extremely cautious attitudes towards investment. It is possible to hedge external risks by reducing R&D expenditure [4] and reducing investment [8]. Bloom found that based on the theory of physical options, due to the characteristics of large R&D investment, long cycle, and high risk, with the increase in policy uncertainty, enterprises will be more cautious about R&D investment [15].

### 2.1.2. Growth Option Theory

The growth option theory is usually used to explain the formation of the Internet bubble in the United States from the end of the 20th century to the beginning of the 21st century [16]. The core is the comparison of costs and benefits. According to this theory, the economic system is full of uncertainty, but this uncertainty will promote investment, thereby promoting economic growth. For industries such as the Internet, the biggest loss of a company's investment is their cost, but once the investment is successful, the company's return is several times their cost. The temptation of such high profits increases speculative investment. Since it takes a period of time for the investment to be converted into production capacity, this investment can be regarded as a "call option" purchased by the enterprise. Bar-Ilan and Strange found empirical evidence for growth options [17]. They believed that for some industries, increased uncertainty will greatly increase expected returns. Atanssov used the election of the governor of the United States to measure policy uncertainty [5]. Through research, it was found that policy uncertainty has a positive impact on corporate innovation, especially in industries that are politically sensitive and difficult to innovate. The promotion effect is particularly obvious. Kraft et al. found that growth options are very important to explain the investment behavior of innovation-driven companies [18]. They found that when uncertainty increases, this type of company will increase R&D expenditures and enable enterprises to obtain high returns in the future. Meng and Shi studied the relationship between economic policy uncertainty and enterprise R&D investment in the DSGE model, and found that the R&D investment of enterprises is positively related to economic policy uncertainty, and the higher the risk preference for enterprises, the more economic policy uncertainty plays a role in promoting R&D investment obviously [6]. Gu et al. distinguished the selection effect and incentive effect of economic policy uncertainty on enterprise innovation and found that economic policy uncertainty is affecting the R&D investment and patent application volume of listed companies, which is different for enterprises with different property rights or industries [7]. Rao et al. found that in times of high uncertainty, enterprises will take more attention to market factors, thus improving their investment efficiency [19].

### 2.1.3. Risk Compensation Theory

In economics, investors need to obtain compensation for risk taking through a risk premium. High uncertainty tends to increase the risk premium, which will further increase the cost of financing. In recent years, a large number of theoretical literature has proved that the increase in uncertainty will increase borrowing costs, intensify the degree of financing constraints of enterprises, and thus affect economic growth [20,21]. Ilut and Schneider proposed the "Ambiguous Business Cycle" model. They defined a group of institutions that were highly uncertain about the future and found that when the uncertainty increased, these institutions reduced their investment and consumption, which in turn affected economic growth [22].

### 2.2. The Impact of Economic Policy Uncertainty on Corporate Green Innovation

Scholars hold the opposite attitude towards the impact of economic policy uncertainty on corporate green innovation. The first is suppression theory. Some scholars believe that the increase in economic policy uncertainty will reduce enterprises' green innovation motivation and innovation ability [2,3,14]. Julio and Yook used official exchanges as a

proxy variable for economic policy uncertainty [23]. The study found that economic policy uncertainty affected business operations and decision making, thereby inhibiting corporate R&D investment. Bhattacharya et al. used patent volume to measure innovation activities in their research, and national election time to measure policy uncertainty periods [4]. They conducted research on data from 43 countries and the results showed that the number of patented inventions increased with the increase in policy uncertainty and policy uncertainty has an inhibitory effect on innovation. Li used the economic policy uncertainty index researched by Baker et al. to measure economic policy uncertainty [24]. Through research, it was concluded that the increase in economic policy uncertainty will inhibit corporate R&D investment. Tan and Zhang, through their empirical research, adopted the belief that economic policy uncertainty will inhibit corporate R&D investment through two transmission mechanisms: real options and financial friction [25].

However, some scholars believed that improving corporate innovation capabilities can enhance corporate core competitiveness and alleviate the negative impact of economic policy uncertainty. Therefore, economic policy uncertainty can encourage companies to innovate and enhance corporate innovation capabilities. Atanssov used the election of the governor of the United States to measure policy uncertainty. Through research, it was found that policy uncertainty has a positive impact on corporate innovation, especially in industries that are politically sensitive and difficult to innovate [5]. The promotion effect is particularly obvious. Meng and Shi conducted an empirical study based on the data of listed companies in China from 2009 to 2015 [6]. The results showed that policy uncertainty has a positive effect on corporate innovation, and policy uncertainty will promote enterprises to seek their own development. Gu et al. studied the incentive effect and selection effect of policy uncertainty on enterprises based on the data of listed companies in China [7]. The study found that policy uncertainty can positively affect the innovation input and innovation output of enterprises. The relationship is affected by factors such as the characteristics of the company's industry and government subsidies.

Some authors believed that economic policy uncertainty and enterprise innovation are jointly affected by "promotional effects" and "inhibition effects". Liu and Huang examined panel data of listed companies in China's strategic emerging industries from 2013 to 2018 and found that there is a U-shaped relationship between economic policy uncertainty and corporate innovation capabilities, and government innovation preferences have a negative regulatory effect [26].

### 2.3. Literature Summary

Throughout the existing literature, scholars at home and abroad have conducted in-depth research on the influencing factors of corporate innovation capabilities and the impact of economic policy uncertainty, forming a wealth of research results. However, the research on how the uncertainty of economic policy affects the green innovation capability of enterprises started relatively late. The following problems exist.

#### 2.3.1. Inconsistent Research Conclusions

On the relationship between economic policy uncertainty and corporate green innovation, scholars have reached inconsistent conclusions. Scholars who hold positive opinions believe that economic policy uncertainty will encourage enterprises to carry out green innovation and alleviate the negative impact of changes in the external environment. Scholars who hold negative opinions believe that the increase in economic policy uncertainty will make the macroeconomic situation unpredictable, delaying the R&D and innovation decisions of business operators and inhibiting the green innovation of enterprises. Some scholars believe that the uncertainty of economic policy is a U-shaped relationship under the combined effect of promotion and inhibition.

### 2.3.2. Few Studies on The Influencing Mechanism

There are relatively few studies on the influence mechanism between economic policy uncertainty and corporate green innovation. Few scholars have paid attention to the transmission path and influence mechanism of the impact of economic policy uncertainty on enterprises' green innovation from a micro perspective. Enterprise innovation capability is the result of the combined effect of micro factors such as enterprise investment decision making and financing scale. Therefore, from a micro perspective, it is an important way to clarify the impact mechanism between economic policy uncertainty and the enterprise of green innovation by studying the impact of increased uncertainty on the choice of investment behavior and financing constraints of enterprises, and then on the innovation ability of enterprises.

Therefore, on the basis of previous studies, this article deeply researches the impact of economic policy uncertainty on the company's green innovation capability, explores its impact mechanism from two aspects of corporate investment and financing, and enriches the research in this field.

### 3. The Impact of Economic Policy Uncertainty on Corporate Green Innovation

First of all, when the uncertainty of economic policy is within a reasonable range, it will promote the improvement of enterprise innovation capabilities. (1) Corporate innovation activities can be regarded as an option. Growth options believe that corporate investment not only focuses on the short-term benefits that investment behavior brings to the company but also pays more attention to the long-term development of the company. Innovation is an important support for enhancing the competitiveness of an enterprise and an important source of excess profits for an enterprise. When the uncertainty of economic policy rises, the uncertainty of the market environment companies faced increases. The frequency of corporate value fluctuations also increases, and corporate value is likely to decline. In this case, many companies tend to take the lead in making innovative investments, seizing market share, and enhancing competitiveness. (2) From the perspective of corporate funds, when policy uncertainty rises, the external operating environment of the company faces greater uncertainty. Due to the savings prevention mechanism, companies tend to reduce their holdings of financial assets and store more cash. At the same time, in order to enhance competitiveness, companies often use part of their funds for innovative research and development, and their innovation capabilities also increase. Therefore, the increase in economic policy uncertainty will increase the market risks faced by enterprises, and enterprises will often increase their investment in innovation, enhance their innovation capabilities, quickly occupy the market, and enhance their competitiveness. Uncertainty in economic policies will enhance corporate innovation capabilities.

However, when economic policy uncertainty continues to rise to a certain level, it will have a restraining effect on corporate innovation. (1) Based on the theory of real options, from the perspective of waiting for options, when the uncertainty of economic policy continues to increase, the uncertainty of the external environment will rise sharply, leading to an increase in the waiting value of options. As time goes by, more and more effective information will be presented, and companies will make decisions to delay investment. Due to the existence of investment irreversibility and principal-agent problems, when the economic policy uncertainty is high, the external environment of the enterprise is confusing and it is difficult for enterprise managers to predict the future development of the enterprise. For the purpose of maximizing their own interests, managers will choose to reduce innovative investment and often invest funds in projects that are most beneficial to them. (2) Sufficient funds are an important guarantee for innovative activities. When economic policy uncertainties continue to increase, financial market frictions and the risk of external capital providers of enterprises increase, which in turn, will cause corporate financing costs to increase. When the economic policy is uncertain, banks cannot accurately judge the risk and return of credit in the market, so they will choose a more conservative credit policy. These unfavorable factors will increase the financing constraints of enterprises

and cause enterprises to reduce their investment in innovation, which has a restraining effect on innovation. (3) As economic policy uncertainties increase, market risks increase and corporate innovation activities themselves have certain risks. The superposition of risks will bring more adverse effects to companies, and some companies will choose to reduce their innovation investment. Therefore, too high uncertainty in economic policies will also inhibit corporate green innovation.

To sum up, economic policy uncertainty not only stimulates corporate R&D investment to promote corporate innovation capabilities, but also inhibits corporate green innovation due to risks brought about by policy uncertainty. The relationship between the uncertainty of economic policy and the green innovation of enterprises under the combined effect of the suppression mechanism and the promotion mechanism may be nonlinear, as shown in Figure 2. When the level of economic policy uncertainty is low, the business environment fluctuations and financing constraints faced by enterprises are relatively small, and the incentive effect brought by policy uncertainty to enterprises will offset the adverse effects brought about by policy uncertainty. The uncertainty of economic policy promotes corporate innovation. When economic policy uncertainty further rises, market risks and financing constraints faced by enterprises further increase. As economic policy uncertainty increases, enterprise innovation will be inhibited. There is an inverted U-shaped relationship between economic policy uncertainty and enterprise innovation ability, which promotes first and then inhibits.

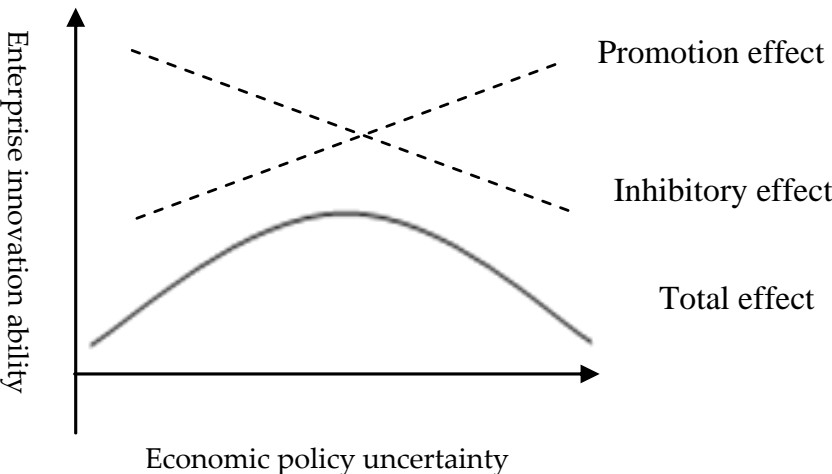

**Figure 2.** The Relationship Between Economic Policy Uncertainty and Corporate Green Innovation.

**4. Research Design on the Impact of Economic Policy Uncertainty on Corporate Green Innovation**

*4.1. Sample Selection and Data Sources*

This paper selects the financial data of China's A-share-listed companies from 2008 to 2019 as a research sample to study the impact of economic policy uncertainty on corporate green innovation. The following processing is performed on the original data: (1) exclusion of the sample of enterprises' with special processing, such as ST and ST* during the observation period. The annual profit of such enterprises is continuously lost and is likely to have the risk of delisting. Such abnormal financial data of enterprises will have an impact on the results of the empirical inspection. (2) Exclusion of the samples of companies that have been PT during the observation period. This type of company's annual net profit has been lost for three consecutive years and the stock will be suspended. The abnormal financial data of this type of company will have an impact on the results of the empirical test. (3) Exclusion of the sample of companies in the financial and insurance industries. The profit model of this type of company is different from that of other industries, and the reflection of corporate innovation capabilities is different from that of ordinary companies. The financial data of this type of company affects the accuracy of the empirical results.

(4) Exclusion of samples of companies with missing important financial data. The financial data of such companies affect the accuracy of the empirical results. In the end, 2356 enterprise samples and 12,141 pieces of data were obtained. Corporate financial data was mainly derived from the Wind database and the CSMAR database, and was supplemented by websites such as Sina Finance and Juchao Information. In order to reduce the influence of outliers, this paper winsorizes 1% on all variables, and in order to alleviate endogeneity, the explanatory variables and all control variables are treated with one period lagging behind.

*4.2. Variable Selection*

4.2.1. Explained Variables

Corporate green innovation (GI) is measured by the logarithm of the number of corporate green patent applications plus one. Considering that the R&D activities of enterprises have a high degree of uncertainty, investment does not necessarily mean that there is output. Compared with R&D investment, innovation output more intuitively reflects the innovation ability of an enterprise. Because the use of green patent indicators can effectively eliminate the impact of other factors (such as innovation subsidies and other policies) on enterprise innovation besides environmental regulatory policies [27], and because of the time lag in the patent authorization process and patents under the influence of human factors such as the preference of authorized examiners, the number of patent applications can truly reflect the innovation level of enterprises more than the number of applications granted. Therefore, in this paper, the number of green patent applications by enterprises plus one is taken as a logarithm to reflect the level of green innovation of enterprises.

4.2.2. Explanatory Variables

Economic policy uncertainty (EPU) currently focuses on three methods. One is based on the measurement of news media and network information. Choosing influential newspapers, per month for each paper, we conducted a search using search words related to economy and policy uncertainty according to media reports in the word frequency statistics, as well as to the expected changes to the tax provisions of macroeconomic prediction deviation of three indicators to measure the uncertainty of economic policy and standardized processing and build up economic policy uncertainty index [1,28,29]. The second is to measure whether there is uncertainty based on whether there is a change of top government officials [2,3]. The third is to consider if the uncertainty of macroeconomic and financial markets is likely to produce a wide range of influence; some scholars will predict variability or volatility as the uncertainty, using the volatility of economic and financial variables, the changes in stock market returns, or profits level of cross-sectional discrete degree of uncertainty as the proxy variable to indirectly measure economic policy uncertainty [30,31].

Volatility of economic and financial variables or cross-sectional dispersion as a measure of economic policy uncertainty is easy to operate and calculate, but it contains non-macroeconomic fundamentals such as risk aversion, leverage effect, and inter-firm heterogeneity. Even if economic policies do not change, these non-macroeconomic fundamentals may still lead to changes in the volatility or cross-sectional dispersion of economic and financial variables. Therefore, in this sense, conditional volatility or dispersion is not equal to uncertainty. Therefore, this paper adopts two methods of news media, network information and official communication, respectively, to measure economic policy uncertainty from national and local levels. The economic policy uncertainty index constructed by news media and network information is used to measure the uncertainty of national economic policy, and the index constructed by local government officials' communication is used to measure the uncertainty of economic policy in prefecture-level cities.

1.  Measure Economic Policy Uncertainty by News Media and Network Information.

The economic policy uncertainty index constructed by Baker et al. is adopted to measure the uncertainty of economic policy [1]. The index, jointly published by Stanford University and the University of Chicago, covers a number of countries around the world. It

measures the uncertainty of economic policies of various countries based on representative media and keyword statistics, and is published on the official website of Economic Policy Uncertainty. The index has good continuity and time variability and has been widely recognized by domestic scholars and abroad. The Chinese economic policy uncertainty index is based on the frequency of certain keywords in articles of the newspaper *The Hong Kong South China Morning Post*. Since the economic policy uncertainty index publishes monthly data, this paper calculates the arithmetic average of monthly data to obtain the annual data and divides the annual data by 100 to keep the order of magnitude consistent. Finally, the economic policy uncertainty index EPU1 is obtained.

2.  Measure Economic Policy Uncertainty by the Exchange of Local Officials

As disseminators and implementers of national policies, local officials' social networks have a subtle effect on their behaviors and decisions, and influence the formulation of local economic policies. According to the definition of economic policy uncertainty (EPU), economic subjects cannot predict whether, when, and how the government will change economic policies [8]. Not only will frequent change in local officials affect the uncertainty of economic policy, but the influence of the transfer of relationships of local officials will also affect the uncertainty of economic policy. This factor is considered in this paper because of the persistence of local officials' cognition of things and management ideas. If an official is transferred from place A to place B, the economic subject in place B can judge how the government changes economic policies based on the management means and governance ideas of the official in place A, and then determine the uncertainty of economic policies.

Therefore, based on the complex network, this paper constructs the local officials' communication network to measure the uncertainty of local economic policy. In the exchange network of official places, the node is a certain place in the network. The side is the transfer relationship between the two places, and the direction of the side is the transfer direction of the official. For example, if the official is transferred from A to B, then the edge points from A to B. The weight of the edge depends on the number of official transfers between the two places and the level of urban development. Because officials conduct their roles not only in the local government but in many other social roles, such as industry association leadership and university professors, this paper defined officials for communication in the network edge as referring to officials in the office who are transferred as different from officials of other forms of flow (such as communication between universities).

In an official communication network in the same province, a clique can be observed, and communication between adjacent provinces may not establish the edge. There will always be a transfer of officials from one province to another, which creates a degree of connection with an office that does not have a border of its own. This creates a weak relationship (as shown in Figure 3). There are enough information advantages and control advantages in the whole network, so that the communication network of the official's office has the rationality of existence.

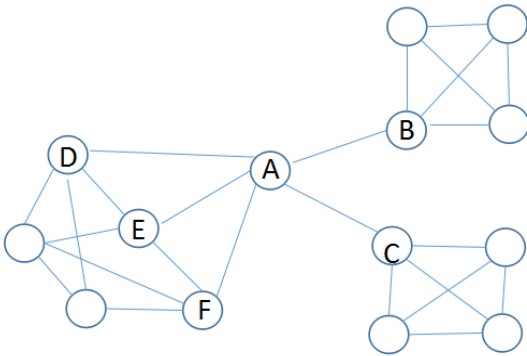

**Figure 3.** Schematic Diagram of Weak Connection Relation.

From Figure 3, the entire network is divided into three community structures; the two arms of AB and AC are the key to the whole network edge and are also important links between the three community structures as A, B, and C have weak relations. The network can provide other offices with a community structure of the internal office without access to official economic policy or experience. Among them, place A is a significant area connecting the entire network. If place A does not exist, the entire network will be disconnected. Therefore, the weak relationship of A is stronger. The transfer of officials through this place can connect the entire network, and there are many channels for obtaining information, which affects the transfer of officials and information transmission of the entire network. In addition, location E is located in the center of the community structure, with a high clustering coefficient and a relatively strong relationship. Officials who pass through the location are transferred more frequently, and economic policies change more frequently.

On the basis of sorting out the resumes of local officials, this paper lists the relationship between the places where Chinese officials hold posts formed during the process of performing their duties. Firstly, from the website of the Central People's Government of the People's Republic of China (home page—Overview of China—Personnel change query), a list of the names of the party secretaries of 343 cities at the prefecture level and above in China from 2008 to 2019 was obtained. The resumes of the officials were found by using the database of local government officials of *People's Daily* online as basic information. Secondly, in order to ensure the authenticity of the communication network relations between the officials and where they work, all the officials with the same name are screened to distinguish whether they are the same person, and each official is given a unique ID code. Again, according to officials' resumes for each successive city list and the municipal party committee secretary of the vice mayor of the above positions of transfer to build the mold "place of employment—place of employment" matrix, the matrix is outlined based on the official flow coupling relationship between different offices. If officials from office i transferred to office j, then the value of matrix (i, j) is 1. Otherwise, the value is 0, and the diagonal between the office and itself is 0.

Centrality analysis in complex networks can quantify the importance of nodes in the network through measurement indexes. Quantitative tools commonly used in complex network analysis are as follows: degree centrality, betweenness centrality, closeness centrality, and eigenvector centrality [32]. By referring to these four commonly used network measurement indicators, we can calculate the closeness of the entire network and the importance of each prefecture in the network. The higher the importance is, the more frequent the exchanges between the officials of the prefecture-level city are, and the higher the uncertainty level of economic policy is.

The economic meaning and the calculation process of the four indicators are as follows:

1.  Degree Centrality Index

Degree centrality is the most direct metric to describe node centrality in network analysis [33]. The higher the node degree of a node is, the higher the degree centrality of the node is, and the more important the node is in the network. It also means that there are frequent changes of officials in the city. The degree centrality index $Degree_i$ is calculated by the following formula.

$$Degree_i = \frac{\sum\limits_j X_{ji}}{(g-1)} \tag{1}$$

The $i$ is a single place of office. The $j$ is a place of office other than the $i$ of the place of office. If there is at least one exchange of officials between the $i$ and the $j$ of office, the $X_{ji}$ is 1. Otherwise, the $X_{ji}$ is 0. The index g is the total number of places of service, using $(g-1)$ to eliminate scale differences.

2.  Betweenness Centrality Index

Betweenness centrality, also known as intermediate centrality, refers to the number of times that the current point serves as the short-circuit bridge between the other



two nodes [34]. In this paper, the number of shortest bridges undertaken by a node is divided by the number of all paths, so as to standardize the data processing. The higher the number of times a node acts as a "bridge", the greater its mediation centrality. It also means that the main leaders of this city are transferred to, or transferred from, other cities frequently. The betweenness centrality index $Betweenness_i$ is calculated by the following formula.

$$Betweenness_i = \frac{\sum\limits_{j<k} g_{jk(n_i)} / g_{jk}}{(g-1)(g-2)/2} \tag{2}$$

The $g_{jk}$ is the number of shortcuts that must be taken to connect the $j$ and $k$ of the place of service. The $g_{jk(ni)}$ is the number of $i$ in the shortcut path to the $j$ and $k$ of the office.

3. Closeness Centrality Index

Closeness centrality reflects the closeness degree between a node and other nodes in the network, which is to take a reciprocal of the shortest distance between the current point and all other nodes [35]. The closer the distance between this point and all other nodes in the network, the greater the closeness centrality and the more it is at the core position of the network. This also means that the city officials have a high degree of communication. The closeness centrality index $Closeness_i$ is calculated by the following formula.

$$Closeness_i = \frac{g-1}{\sum\limits_{j=1}^{g} d(i,j)} \tag{3}$$

The $d(i, j)$ is the distance from the $i$ to the $j$ of the post in the network. If a given place is not linked to all places of service, the proximity centrality cannot be accurately calculated through the non-complete relationship. The sum of the number of places directly connected to the place of service is divided and then multiplied by its proportion in the total number of places on the network.

4. Eigenvector Centrality Index

Eigenvector centrality is the normalized eigenvector corresponding to the maximum eigenvalue of the adjacency matrix [35]. The greater the centrality of the eigenvector is, the more important the neighbor of the node is, and it is an indication to measure the importance of its neighbor. The eigenvector centrality index $Eigenvector_i$ is calculated by the following formula.

$$Eigenvector_i = \frac{1}{\lambda} \sum_j b_{ij} E_j \tag{4}$$

The $b_{ij}$ is the adjacency matrix and there is at least one official exchange between the $i$ and the $j$ of the post. Then, the $b_{ij}$ is 1, otherwise it is 0. The $\lambda$ is the maximum eigenvector of the B, and $E_j$ is the characteristic value of the $j$ centrality of the place of service.

### 4.2.3. Control Variables

Based on the existing literature, important indicators affecting green innovation of enterprises are selected as follows: firm size (Size), corporate profitability index return on net assets (LEV), corporate debt structure index asset–liability ratio (ROE), enterprise asset structure index tangible assets' ratio (Tang), the ratio of cash flow enterprise cash flow status indicators (Cash), and enterprise equity structure index (OC) equity concentration as control variable. Both individual effects and time effects were controlled. Variable definitions and calculation methods are listed in Table A1 which placed in the Appendix A.

### 4.3. Model Design

In order to test the relationship between economic policy uncertainty and the enterprise of green innovation, this paper first adopts the Hausman test to verify that fixed effect

regression should be adopted. Secondly, the likelihood-ratio test was used to prove the existence of the time effect. Finally, considering that the text panel data are in the form of a large sample and consist of a short time period, the White test is adopted to verify the existence of heteroscedasticity in data so a robust standard error is adopted for regression. The fixed effect regression model is as follows.

$$GI_{i,t+1} = \beta_0 + \beta_1 EPU_{i,t} + \beta_2 EPU^2_{i,t} + Control_{i,t} + year + \varepsilon \tag{5}$$

Enterprises' innovation activities are characterized by long-term nature. In order to alleviate the endogenous problem, all explanatory variables and control variables are lagged behind the explained variables by one period. Where, *i* represents a listed company, and *t* represents the corresponding year. $GI_{i,t+1}$ represent the innovation capability of company *i* in *t + 1* year. $EPU_t$ represents the uncertainty degree of economic policy in *t*. *Control* represents a series of control variables. *year* is the fixed effect of the year, and $\varepsilon$ is the random disturbance term of the fixed-effect model. If $\beta_1$ is significantly positive and $\beta_2$ is significantly negative, there is an inverted U-shaped relationship between economic policy uncertainty and firm green innovation. If $\beta_1$ is significantly negative and $\beta_2$ is significantly positive, there is a positive U-shaped relationship between economic policy uncertainty and firm green innovation.

## 5. Empirical Test and Result Analysis

### 5.1. Descriptive Statistical Analysis of Variables

Table 1 reports the results of a descriptive statistical analysis of the main variables used in the sample study. The data analysis results show that the mean value of green innovation capability (GI) of enterprises is 2.80, the median value is 2.83, the maximum value is 6.87, and the minimum value is 0. The data distribution is uneven, and the gap between the maximum value and minimum value is large, indicating that there is a big gap in green innovation capability and innovation effect among enterprises. The standard deviation of the economic policy uncertainty index (EPU1) was 1.73, the minimum was 0.99, and the maximum was 7.92. The standard deviation of EPU2 is 0.867, the minimum value is 0.006, and the maximum value is 1.956, indicating that the economic policy of the study sample fluctuates greatly. Table 1 also reports descriptive statistical results of enterprise size (Size), return on equity (ROE), asset–liability ratio (LEV), tangible asset ratio (Tang), cash flow ratio (Cash), and equity concentration ratio (OC).

**Table 1.** Descriptive Statistics.

| Variable | N | Median | Mean | Standard Deviation | Minimum | Maximum |
|---|---|---|---|---|---|---|
| GI | 12,141 | 2.830 | 2.800 | 1.620 | 0 | 6.870 |
| EPU1 | 12,141 | 1.810 | 2.730 | 1.730 | 0.989 | 7.919 |
| EPU2 | 12,141 | 0.915 | 0.867 | 0.517 | 0.006 | 1.956 |
| Size | 12,141 | 21.90 | 22.09 | 1.260 | 19.99 | 26.03 |
| ROE | 12,141 | 0.070 | 0.070 | 0.090 | −0.350 | 0.330 |
| LEV | 12,141 | 0.390 | 0.400 | 0.200 | 0.0500 | 0.850 |
| OC | 12,141 | 0.330 | 0.350 | 0.150 | 0.0900 | 0.740 |
| Tang | 12,141 | 0.660 | 0.650 | 0.200 | 0.150 | 0.990 |
| Cash | 12,141 | −0.0200 | −0.0400 | 0.160 | −0.650 | 0.370 |

The paper also conducted a Pearson correlation test and a multicollinearity test, and found that there was a significant positive correlation between firms' green innovation (GI) and economic policy uncertainty (EPU). Whether there was a simple linear correlation or an inverted U-shaped relationship between firms' green innovation capability and economic policy uncertainty remains to be further tested. The maximum value of variance inflation factor (VIF) is 3.03, far less than 10, indicating that the model does not have a complete multicollinearity problem.

### 5.2. Analysis of Regression Results

Table 2 reports the regression results of the impact of economic policy uncertainty, EPU, on the enterprise of green innovation GI after controlling the fixed effect of the year. Column (1) is the regression result of the impact of economic policy uncertainty EPU1 on enterprise green innovation GI when measuring economic policy uncertainty according to the Baker index. Column (2) is a local officials' communication network constructed by a complex network model to measure the uncertainty of economic policy and obtain the regression result of the impact of economic policy uncertainty EPU2 on enterprise green innovation GI. The results of both empirical tests show that the regression coefficient of the primary term of EPU is significantly positive at the 1% level, and the regression coefficient of the second term of EPU is significantly negative at the 1% level, indicating that there is an inverted U-shaped relationship between economic policy uncertainty, EPU, and enterprise green innovation, GI. In the initial stage, the increase in economic policy uncertainty will promote the green innovation of enterprises, but the high uncertainty of economic policy will inhibit the green innovation of enterprises.

**Table 2.** Regression Analysis Results.

| Variables | (1) GI | (2) GI |
|---|---|---|
| EPU1 | 0.940 *** (11.04) | |
| EPU1$^2$ | −0.073 *** (−10.46) | |
| EPU2 | | 0.020 *** (3.485) |
| EPU2$^2$ | | −0.140 *** (−14.370) |
| Size | −0.718 *** (−5.34) | −0.009 *** (−2.684) |
| LEV | 0.104 (0.76) | −0.087 *** (−2.724) |
| OC | −0.245 (−1.03) | −0.003 *** (−12.071) |
| Tang | −0.047 (−0.42) | −0.002 *** (−4.179) |
| Cash | 0.151 *** (2.58) | −0.443 *** (−22.379) |
| Constant | −11.839 *** (−12.75) | 0.319 *** (4.306) |
| Year | Yes | Yes |
| N | 12,141 | 12,141 |
| F | 53.847 | 276.921 |
| R$^2$ | 0.295 | 0.168 |

Note: *t* statistics in parentheses, *** represents $p < 0.001$.

A Utest test is further conducted to verify the inverted U-shaped relationship between economic policy uncertainty and enterprise green innovation, and the critical value of the impact of economic policy uncertainty on enterprise green innovation is determined. The results show that the *p* value is less than 0.05, which rejects the null hypothesis and verifies the inverted U-shaped relationship between economic policy uncertainty and enterprise green innovation. The critical value of the national economic policy uncertainty index EPU1 is 6.413, and the maximum value of EPU1 is 7.719; the minimum value of EPU1 is 0.989, and the critical value 6.413 is within the value range of 0.989–7.919. This indicates that the uncertainty of China's economic policy from 2008 to 2019 is distributed on both sides of the critical value. In some years, the economic policy changes are too frequent, and the uncertainty is high, which inhibits the green innovation of enterprises. The critical value obtained by the Utest test has practical significance. When EPU1 is less than 6.413, the

increase in economic policy uncertainty will promote enterprises' green innovation; when EPU1 is higher than 6.413, the increase in economic policy uncertainty will inhibit enterprises' green innovation. When the economic policy uncertainty index EPU1 reaches 6.413, it has the strongest promoting effect on the green innovation of enterprises. The mean value of EPU1 is 2.730, which is far less than the inflection point of 6.413, indicating that the uncertainty of national economic policy is on the left side of the inverted U-shaped curve in most years in China, which can promote the improvement of enterprises' green innovation ability.

At the regional level, the inflection point of EPU2 is 0.071. The maximum value of EPU2 is 1.956, and the minimum value of EPU2 is 0.006. The inflection point 0.071 is within the value range of 0.006–1.956. The mean value of EPU2 is 0.867, which is higher than the inflection point 0.071, indicating that the uncertainty of economic policies in most regions of China is on the right side of the inverted U-shaped curve, and the change of regional economic policies is too frequent, which inhibits the improvement of enterprises' green innovation ability. Therefore, at present, China's national macroeconomic policy has a more obvious role in promoting enterprise green innovation. Local governments should improve the stability of economic policies and reduce frequent changes in economic policies.

*5.3. Robustness Test*

In order to ensure the robustness of the empirical test results, the following robustness analysis is conducted in this paper.

5.3.1. Re-Calculation of Economic Policy Uncertainty Index

In the empirical test of this paper, the economic policy uncertainty index (EPU1) is converted into annual data by means of arithmetic average of monthly data. Based on the research of Gu [7], this paper converts the geometric average of monthly data into annual data and repeats the above empirical research process. The regression results are shown in Table 3. The results show that the primary term of EPU index is significantly positive at the level of 1%, and the second term of EPU index is significantly negative at the level of 1%, which again verifies the inverted U-shaped relationship between economic policy uncertainty and the enterprise of green innovation.

**Table 3.** Robustness Test Results.

| Variables | (1) GI | (2) GI |
|---|---|---|
| EPU | 0.231 *** | 1.103 *** |
|  | (11.64) | (11.19) |
| EPU$^2$ |  | −0.092 *** |
|  |  | (−10.81) |
| Size | 0.560 *** | 0.560 *** |
|  | (12.36) | (12.36) |
| ROE | −0.718 *** | −0.718 *** |
|  | (−5.34) | (−5.34) |
| LEV | 0.104 | 0.104 |
|  | (0.76) | (0.76) |
| OC | −0.245 | −0.245 |
|  | (−1.03) | (−1.03) |
| Tang | −0.047 | −0.047 |
|  | (−0.42) | (−0.42) |
| Cash | 0.151 *** | 0.151 *** |
|  | (2.58) | (2.58) |
| Constant | −10.790 *** | −12.025 *** |
|  | (−11.22) | (−13.02) |
| Year | Yes | Yes |
| N | 12,141 | 12,141 |
| R$^2$ | 0.295 | 0.295 |

Note: *t* statistics in parentheses, *** represents *p* < 0.001.

### 5.3.2. Re-Measurement of Enterprise Innovation Capability Index

This paper interpreted the variable green innovation ability of enterprises by the listed company by taking the green patent applications logarithmic measure. In order to avoid the instability of empirical results caused by missing samples, this article takes the current innovation input of enterprises divided by total assets as the proxy variable of green innovation of enterprises. The above empirical process is repeated and the regression results as shown in Table 4. The empirical test results are consistent with the hypothesis in this paper.

**Table 4.** Analysis of Robust Results.

| Variables | (1) GI | (2) GI |
| --- | --- | --- |
| EPU | 0.003 *** | 0.009 *** |
| | (10.02) | (8.11) |
| $EPU^2$ | | −0.001 *** |
| | | (−7.45) |
| Size | 0.009 *** | 0.009 *** |
| | (5.59) | (5.59) |
| ROE | 0.002 | 0.002 |
| | (1.49) | (1.49) |
| LEV | 0.004 | 0.004 |
| | (1.28) | (1.28) |
| OC | 0.001 *** | 0.001 *** |
| | (4.34) | (4.34) |
| Tang | −0.001 | −0.001 |
| | (−0.74) | (−0.74) |
| Cash | −0.004 *** | −0.004 *** |
| | (−6.06) | (−6.06) |
| Constant | 0.059 *** | 0.049 *** |
| | (5.30) | (4.56) |
| Year | Yes | Yes |
| N | 12,141 | 12,141 |
| $R^2$ | 0.09 | 0.09 |

Note: *t* statistics in parentheses, *** represents $p < 0.001$.

## 6. The Moderating Mechanism Analysis of the Relationship between Economic Policy Uncertainty and Enterprise Green Innovation

This paper further analyzes the moderating mechanism analysis of the relationship between economic policy uncertainty and enterprise green innovation from the perspective of firm behavior choice. First, from the perspective of enterprise financing behavior, according to the theory of financing constraint, innovation behavior is exclusive, and there is often serious information asymmetry between the two sides of the investment, resulting in enterprise innovation often being faced with serious financing constraints. In enterprises with different financing constraints, the impact of economic policy uncertainty on green innovation will be different. Secondly, from the perspective of enterprise investment behavior, under the background of limited enterprise resources, the change of enterprise investment behavior preference will also affect the enterprise of green innovation investment, and then affect the enterprise of green innovation level. Therefore, among firms with different investment behavior preferences, the impact of economic policy uncertainty on green innovation is also different.

### 6.1. Theoretical Analysis of the Moderating Mechanism Affecting the Relationship between Economic Policy Uncertainty and Enterprise of Green Innovation

6.1.1. Impact of Financing Constraints

Innovation requires significant and stable financial support [36]. Internal financing alone can hardly meet the financial needs of innovation, and external financing has become

an important source of innovation funding for enterprises [37]. Therefore, the availability of external financing becomes an essential constraint for enterprises to engage in innovative R&D activities. Three factors contribute to the high cost of green innovation financing and the difficulty of obtaining external financing for enterprises. First, innovation activities are characterized by high capital requirements, long payback periods, uncertainty, and high innovation risk, leading banks and credit investors to be reluctant to invest capital in innovation activities [38]. Second, in order to avoid competitors learning their core secrets, companies are reluctant to disclose detailed information about their innovation activities, resulting in a serious information asymmetry between companies and external investors [39,40]. Finally, the results formed by innovation outputs are basically intangible assets, which are not easily collateralized, and thus R&D enterprises cannot easily obtain bank loans [41]. As a result, corporations face a serious funding gap in innovation [38], high external financing costs [39], and obvious financing constraints [42]. Unstable funding sources can easily lead to interruptions in firms' innovation activities and constrain their independent R&D [43]. Therefore, financing constraint is an important variable affecting the enterprise of green innovation.

With the increase in economic policy uncertainty, market fluctuations and market risks increase, and enterprises face more uncertainty in the operating environment. As a result, the investment risks of external capital providers such as capital market and venture investors increase, so the cost of external financing increases [44]. At the same time, the rising uncertainty of economic policies will aggravate the credit risks of banks, and banks and credit departments will adopt relatively conservative credit policies [45]. Banks and other credit departments will increase their examination of enterprises' loan qualification and solvency, making it more difficult for enterprises to obtain loans from banks, and subsequently, the number of bank loans will decrease. Therefore, rising economic policy uncertainty will increase financing difficulties for enterprises.

At this point, enterprises with low financing constraints and abundant capital are more likely to alleviate the impact of economic policy fluctuations on the lack of green innovation funds. However, enterprises with higher financing constraints and capital shortage will be more conservative in the face of the increased uncertainty of the external economic environment caused by the fluctuations of economic policies [46] and reduce the speed of external expansion and investment in green innovation. Therefore, financial constraint difference is an important moderating factor influencing the relationship between economic policy uncertainty and green innovation of enterprises.

### 6.1.2. The Impact of Corporate Investment Behavior Choices

Capital is profit seeking. In order to obtain high returns from investment in real estate and financial assets, there is an economic phenomenon that enterprises invest a lot of resources in real estate and financial assets. This economic phenomenon reflects the changes in the investment behavior of enterprises and is called the financialization of enterprises. In the context of limited resources, with the increase in the financial degree of enterprises, financialization will have a "crowding out effect" on real investment [47]. Excessive holdings of financial assets will lead firms to divert from their primary business and focus too much on the short-term benefits of financial assets. In the context of limited enterprise resources, investment in financial assets will reduce investment in innovation and weaken the foundation of manufacturing development [48], leading to the gradual deviation of real enterprises from their main business and forming the phenomenon of "hollowing out of manufacturing" [49], which makes enterprises lack sufficient funds. This makes enterprises lack sufficient funds for equipment renovation and product R&D innovation, which weakens their innovation capacity [50,51]. Therefore, financialization is also an important variable affecting the enterprise of green innovation.

With the increase in economic policy uncertainty, enterprises' future income, costs, and cash flow are highly uncertain, making it more difficult for enterprises to raise funds [45]. This will exacerbate the finiteness of enterprise resources. Under the background of a

possible liquidity shortage, enterprises with a higher degree of financialization of investment behavior will put more resources into investment activities such as investment real estate and financial assets, and have a higher degree of resource crowding on green innovation behavior. This will reduce the uncertainty of economic policy for the promotion of green innovation and increase economic policy uncertainty inhibitory effect of green innovation of the enterprise. On the other hand, enterprises with low financialization of investment behaviors focus more on their main business [49] and invest less resources in investment activities such as real estate and financial assets, occupying less resources related to green innovation behaviors. With the increase in economic policy uncertainty, the inhibiting effect of resource shortage on enterprises' green innovation will be weakened, and the negative impact of the capital chain fracture on production and operation activities will be reduced [52]. Therefore, the financialization difference of firm investment behavior is also an important moderating factor affecting the relationship between economic policy uncertainty and firm green innovation. The moderating mechanism affecting the relationship between economic policy uncertainty and enterprise of green innovation is shown in Figure 4.

Degree of Financialization of Enterprises

Economic Policy Uncertainty ———Inverted U——— Corporate Innovation Capability

Financing constraints

**Figure 4.** Research Idea Figure.

*6.2. Empirical Test Model Setting*

6.2.1. Regulating Variable Definition

The main measures of financing constraints are the KZ index [53], the WW index [54], and the SA index [55]. Since the KZ index and the WW index contain endogenous financial variables of firms, they will generate measurement bias. Therefore, this paper uses the SA index as a proxy variable for financing constraints. The SA index is proposed by Hadlock and Pierce and consists of two indicators: firm size and firm age [55]. The calculation formula is as follows:

$$SA = 0.737 * Size - 0.04 * Age + 0.043 * Size^2 \tag{6}$$

The SA index is usually negative, and the absolute value of SA is generally used to measure the degree of corporate financing constraints; the larger the absolute value of SA, the more serious the corporate financing constraints.

Drawing on the studies of Song et al. and Xiao [56,57], this paper selects the aggregate of trading financial assets, available-for-sale financial assets, derivative financial assets, held-to-maturity investments, investment properties, and long-term equity investments as corporate financial assets based on the balance sheet. The degree of financialization (FIN) of an enterprise is measured by the ratio of the enterprise's financial assets to the enterprise's total assets at the end of the period, and the larger the ratio, the higher the degree of financialization of the enterprise.

6.2.2. Model Construction

In order to test the influence of financialization of firm investment behavior and financing constraint on economic policy uncertainty and firm green innovation behavior, this paper divided the samples into different groups. By observing the inflection point shift and flat or steep curve change of the influence curve, we can judge the change of the impact of economic policy uncertainty on green innovation of enterprises with differences in financialization degree and financing constraints [58].

The average value of SA of the financing constraint index is calculated according to the year. The sample whose SA value is higher than the average value of the current year is categorized as the High FC group, and the sample whose SA value is lower than the average value is categorized as the Low FC group. By referring to the fixed effect regression model (5) constructed in 4.3, the two groups of samples are empirically tested to compare the change of the influence curve of economic policy uncertainty on enterprises' green innovation and judge whether financing constraints affect the relationship between economic policy uncertainty and enterprises' green innovation.

Similarly, the average value of the FIN index is obtained by year. The sample whose FIN value is higher than the average value of the current year is categorized as the High FIN group, and the sample whose FIN value is lower than the average value is categorized as the Low FIN group. By referring to the fixed effect regression model (5) constructed in 4.3, the two groups of samples are empirically tested to compare the change of the influence curve of economic policy uncertainty on enterprises' green innovation and judge whether financialization affects the relationship between economic policy uncertainty and enterprises' green innovation.

*6.3. Analysis of Empirical Test Results*

Table 5 reports the test results of the moderating effects of financing constraint and financialization on the relationship between economic policy uncertainty and enterprise green innovation under the control of annual fixed effect. Column (1) is listed as the regression result of the impact of economic policy uncertainty on enterprise green innovation in the sample group with high financing constraints. Column (2) is listed as the regression result of the impact of economic policy uncertainty on green innovation of enterprises in the sample group with low financing constraints. It is found that in the two models, the regression coefficients of economic policy uncertainty EPU1 are all significantly positive at the 1% level, and the regression coefficients of EPU1 quadratic term are all significantly negative at the 1% level. The results indicate that there is an inverted U-shaped relationship between economic policy uncertainty and green innovation in enterprises with high and low financing constraints. The regression models of the influence of economic policy uncertainty on the enterprise of green innovation in two groups of samples were obtained. Formula (7) is the regression model of the impact of economic policy uncertainty on the green innovation of enterprises with high financing constraints. Formula (8) is the regression model of the impact of economic policy uncertainty on the green innovation of enterprises with low financing constraints.

$$GI_{i,t+1} = 0.819 \times EPU_{i,t} - 0.063EPU^2_{i,t} + Control_{i,t} - 11.119 \qquad (7)$$

$$GI_{i,t+1} = 1.033 \times EPU_{i,t} - 0.081EPU^2_{i,t} + Control_{i,t} - 12.422 \qquad (8)$$

According to the regression model, the curve of the impact of economic policy uncertainty on the green innovation of enterprise with high and low financing constraints can be obtained, as shown in Figure 5. The left side of the inverted U-shaped curve is steeper in the low financing constraint firms than in the high financing constraint firms. This indicates that economic policy uncertainty has a more significant promoting effect on green innovation in enterprises with low financing constraints [58]. Therefore, financing constraints can effectively adjust the impact of economic policy uncertainty on green innovation.

**Table 5.** Regression Analysis Results of the Moderating Effects.

| Variables | (1) | (2) | (3) | (4) |
| --- | --- | --- | --- | --- |
| | **High FC** | **Low FC** | **High FIN** | **Low FIN** |
| EPU1 | 0.819 *** | 1.033 *** | 0.742 *** | 1.000 *** |
| | (7.28) | (7.74) | (4.71) | (9.63) |
| $EPU1^2$ | −0.063 *** | −0.081 *** | −0.056 *** | −0.079 *** |
| | (6.75) | (−7.43) | (−4.27) | (−9.12) |
| Size | 0.544 *** | 0.573 *** | 0.582 *** | 0.565 *** |
| | (9.75) | (7.49) | (6.23) | (11.64) |
| ROE | −0.682 *** | −0.674 *** | −0.596 ** | −0.586 *** |
| | (−3.94) | (−3.12) | (−2.38) | (−3.42) |
| LEV | 0.158 | −0.005 | −0.034 | 0.057 |
| | (0.84) | (−0.02) | (−0.13) | (0.34) |
| OC | −0.793 ** | 0.267 | −0.430 | −0.208 |
| | (−2.35) | (0.72) | (−0.81) | (−0.72) |
| Tang | 0.080 | −0.165 | −0.153 | 0.088 |
| | (0.54) | (−0.94) | (−0.77) | (0.66) |
| Cash | 0.203 ** | 0.112 | 0.235 ** | 0.199 *** |
| | (2.51) | (1.30) | (2.34) | (2.63) |
| Constant | −11.119 *** | −12.422 *** | −11.692 *** | −12.198 *** |
| | (−9.17) | (−8.17) | (−5.79) | (−12.42) |
| Year | Yes | Yes | Yes | Yes |
| N | 6069 | 6072 | 3689 | 8452 |
| R-squared | 0.266 | 0.283 | 0.224 | 0.298 |
| F | 40.24 | 47.86 | 17.72 | 71.49 |
| r2_a | 0.263 | 0.281 | 0.220 | 0.296 |

Note: *t* statistics in parentheses, ** represents $p < 0.05$, *** represents $p < 0.001$.

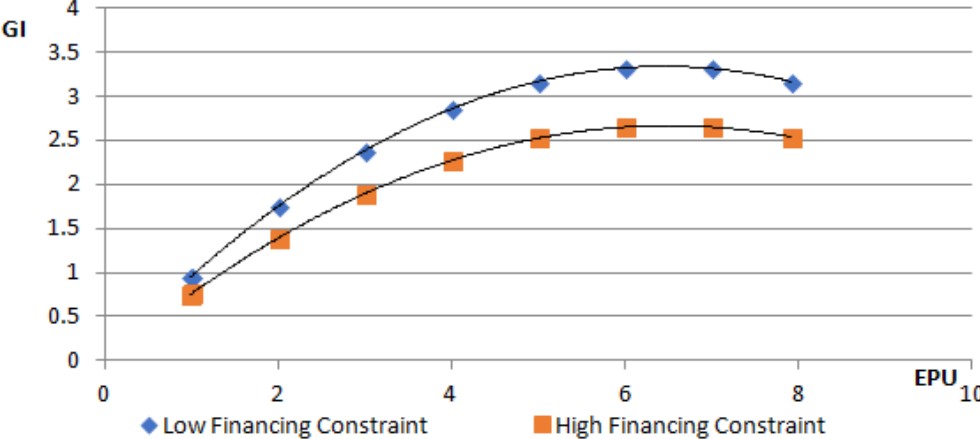

**Figure 5.** The Curve of the Impact of Economic Policy Uncertainty on the Enterprise of Green Innovation.

Column (3) in Table 5 is listed as the regression results of the impact of economic policy uncertainty on the green innovation of enterprises in the high financialization group. Column (4) is listed as the regression result of the impact of economic policy uncertainty on the enterprise of green innovation in the low financialization group. It is found that in the two models, the regression coefficients of economic policy uncertainty EPU1 are all significantly positive at the 1% level, and the regression coefficients of EPU1 quadratic term are all significantly negative at the 1% level. The results indicate that there is an inverted U-shaped relationship between economic policy uncertainty and green innovation in enterprises with high and low financialization. The regression models of the influence of economic policy uncertainty on the enterprise of green innovation in two groups of samples were obtained. Formula (9) is the regression model of the impact of economic policy uncertainty on the green innovation of enterprises with high financialization. Formula (10)

is the regression model of the impact of economic policy uncertainty on the green innovation of enterprises with low financialization.

$$GI_{i,t+1} = 0.742 \times EPU_{i,t} - 0.056EPU^2_{i,t} + Control_{i,t} - 11.692 \qquad (9)$$

$$GI_{i,t+1} = 1.000 \times EPU_{i,t} - 0.079EPU^2_{i,t} + Control_{i,t} - 12.198 \qquad (10)$$

According to the regression model, the curve of the impact of economic policy uncertainty on the green innovation of enterprise with high and low financialization can be obtained, as shown in Figure 6. The left side of the inverted U-shaped curve is steeper in the low financialization firms than in the high financialization firms. It indicates that economic policy uncertainty has a more significant promoting effect on green innovation in enterprises with low financialization [58]. Therefore, financialization can effectively adjust the impact of economic policy uncertainty on green innovation.

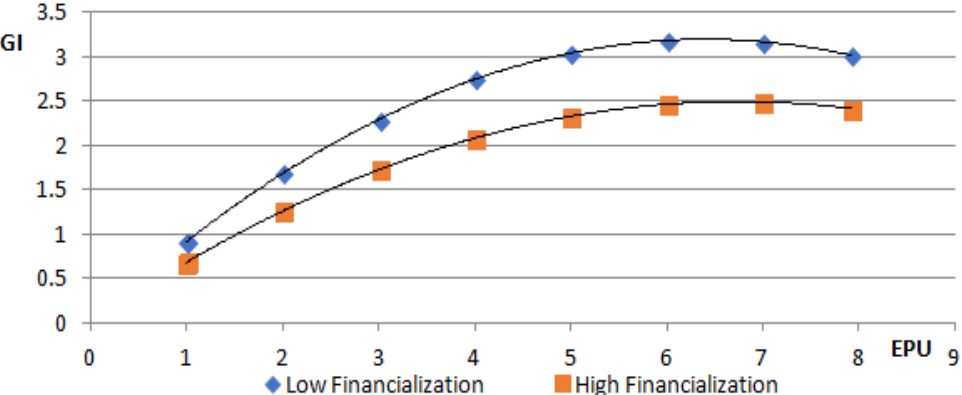

**Figure 6.** The Curve of the Impact of Economic Policy Uncertainty on the Enterprise Green Innovation.

## 7. Conclusions and Recommendations

This paper examined the impact of economic policy uncertainty on green innovation at both national and regional levels. The Baker index based on news media and network information was used to measure the uncertainty of national economic policy, and the official exchange index based on the complex network was used to measure the uncertainty of economic policy in prefecture-level cities. It was found that there is an inverted U-shaped relationship between economic policy uncertainty and firms' green innovation capability. Moreover, the uncertainty index of national macroeconomic policy is mostly on the left side of the inverted U shape, which can promote the improvement of enterprises' green innovation ability. However, too frequent changes in regional economic policies will inhibit enterprises' green innovation ability. This paper further analyzed the moderating effect of financialization of investment behavior and financing constraint on the impact of economic policy uncertainty on the green innovation of enterprises from the perspective of investment and financing behavior choice. It was found that the impact of economic policy uncertainty on green innovation is more obvious for firms with low financing constraints and low financialization.

The research content of this article still has some limitations. In the future, the authors intend to further explore the causes and mechanisms of the impact of economic policy uncertainty on green innovation. This paper finds that the impact of economic policy uncertainty on green innovation presents an inverted U shape. However, there is no further analysis of the reasons for maintaining the inverted U-shaped relationship between them. Macroeconomic policy changes will affect enterprises' choice of green innovation behavior from many aspects. However, it is very difficult to find instruments affecting one of many endogenous variables, but not the other. This also makes it difficult to analyze the reasons and ways that economic policy uncertainty affects the enterprise of green innovation. Of course, this is also the direction and breakthrough point of the author's future research.

Based on the above findings, this paper makes the following recommendations to governments and enterprises.

### 7.1. Suggestions for the Government to Formulate Economic Policies

Government policies are an important tool for macroeconomic regulation and control, and the innovation activities of enterprises are subject to both the "facilitating effect" and the "inhibiting effect" of economic policies, resulting in a non-linear relationship between the policy uncertainty and innovation capability of enterprises. The relationship between policy uncertainty and firms' innovation capacity is non-linear. Therefore, when formulating policies, policymakers should fully consider the issues of gains and losses and costs and benefits. Furthermore, policymakers should consider the incentive effects and negative impacts of policy uncertainty on enterprises' micro behavior and reasonably grasp the frequency and magnitude of policy adjustments.

Additional recommendations include increasing efforts to promote the reform and development of the financial market, reducing financing costs, and broadening financing channels. Enterprise innovation activities need to inject a large amount of capital, and some enterprises still have the problem of expensive and difficult financing. The government should focus on optimizing the financing environment of the financial market, widening enterprise financing channels, reducing enterprise financing costs, and thus alleviating enterprise financing constraints and solving the issue of expensive and difficult financing for enterprises.

Establishing a sound market mechanism further improves the investment environment for enterprises, promotes enterprises to gain from the real economy, and reduces the financialization of enterprises. The sound market mechanism can promote the rational allocation of resources, improve the price mechanism, stimulate the innovation vitality of enterprises through perfect market allocation, prompt enterprises to take the initiative to reasonably apply funds in innovation projects, and give full play to the role of government macro control.

### 7.2. Suggestions for Improving the Green Innovation Capacity of Enterprises

Enterprises should look at policy fluctuations rationally. The increase in economic policy uncertainty is a coexistence of risks and opportunities for enterprises. If enterprises regard policy uncertainty as a single risk, it will affect their innovation activities and hinder their development. If enterprises regard policy uncertainty as an opportunity, they will be hit hard by pursuing profits and blindly carrying out innovation activities. Enterprises need to rationally analyze the opportunities and risks brought by economic policy uncertainty to gain an advantageous position in the competition.

Finally, strengthening the control of enterprise capital is important. Adequate cash flow is the basis for maintaining the daily operation of the enterprise, which helps the enterprise fully grasp opportunities and make flexible decisions, and is conducive to sustainable development. It is particularly important for enterprises to strengthen their own financial control in daily operation, strengthen the control of cash flow, and improve capital management, especially when there is a high degree of uncertainty in economic policies.

**Author Contributions:** Conceptualization, W.Z. and L.C.; methodology, X.H.; formal analysis, H.D. and Y.X.; investigation, Z.W.; data curation, X.H.; writing—original draft preparation, W.Z. and X.H.; writing—review and editing, L.C. and H.D.; funding acquisition, W.Z. All authors have read and agreed to the published version of the manuscript.

**Funding:** This work was supported by the National Social Science Foundation of China, grant number 18BGL185. The work was funded by Wenjun Zhou.

**Institutional Review Board Statement:** Not applicable.

**Informed Consent Statement:** Not applicable.

**Data Availability Statement:** Not applicable.

**Conflicts of Interest:** The authors declare no conflict of interest.

## Appendix A Appendix

**Table A1.** Variable Definitions.

| Variable Types | The Variable Name | Symbol | Variable Calculation Method |
|---|---|---|---|
| Explained variable | Green innovation | GI | Ln(1 + number of green patent applications) |
| Explanatory variables | Economic policy uncertainty index | EPU1 | Baker economic policy uncertainty index |
| | | EPU2 | Local officials communication index based on complex network |
| Moderating variables | Financing constraints | SA | $SA = 0.737 * Size - 0.04Age + 0.043 * Size^2$ |
| | Financialization | FIN | Financial assets/total assets |
| | The enterprise scale | Size | The natural log of total assets at year end |
| | Return on equity | ROE | Net profit/total assets |
| Control variables | Asset–liability ratio | LEV | Total liabilities/total assets |
| | Tangible asset ratio | Tang | Tangible assets/total assets |
| | Cash flow ratio | Cash | (Cash flow from operating activities + cash flow from investing activities)/total assets |
| | Ownership concentration | OC | Shareholding ratio of the largest shareholder |

Source: manual collation by author.

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
