# Peer review of "Research on the Impact of Economic Policy Uncertainty on Enterprises’ Green Innovation—Based on the Perspective of Corporate Investment and Financing Decisions"

_sustainability, doi:10.3390/su14052627_

Round 1

Reviewer 1 Report

Review of the manuscript: Sustainability-1575319

This study examines the relationship between economic policy uncertainty and  companies’ green innovation in China for the period 2008–2019. The authors also investigate the mediating roles of financial constraint and financial assets held in explaining this relationship. The results support the authors’ expectations for inverse U shape in those relationships.

I agree with the authors that the evidence for the potential impact of economic policy uncertainty on  companies’ investment policies in the existing literature is inconclusive. I also think that the authors do a very good job in developing theoretical arguments and documenting supporting empirical evidence. Therefore, this study has a potential to contribute to the related literature. However, I believe that the study needs to be improved by addressing my following concerns below.

Major Concerns:

  1. The set-up of the manuscript is not engaging yet. First, the abstract is too long and not very clear to capture unique points of this study in order to extract the main contribution. Second, the introduction, however, is too short and/or not well designed to accomplish the following important tasks: describe the problem or phenomenon; tell the reader why it is important; identify the gap in the literature; explain how your study will address this gap (including the theoretical lens); explain the contribution in detail. All those issues have been addressed in the sections for the literature review and summary. The authors are expected to design the set-up from the perspective I listed in this paragraph.
  1. The claims for the mediating roles of financial constraints and financialization in section 6, which is about the analysis on the transmission mechanism of enterprise green innovation affected by economic policy uncertainty, have not been grounded well. The empirical analysis in this section is very complex and makes the idea follow very difficult. I don’t think that the three-step approach proposed with standard OLS regressions to test the mediating roles of two mechanisms is not a proper empirical set-up. The idea in this section requires to performs more sophisticated analyses, such as instrumental variable repressions and/or (dynamic) GMM, to capture the simultaneity and/or endogeneity of financial constraint and financialization determined by economic policy uncertainty, which is also a main determinant of green innovation. However, it is very difficult, if not impossible, to find instruments affecting one of two endogenous variables, but not the other. Dynamic GMM may require some certain assumptions and requirements. My suggestion to the authors is to convert the mediating role of financial constraints and financialization to moderating roles, which propose a joint effect of these two factors along with economic policy uncertainty on green innovation. This will generate a more reliable approach to follow the arguments as well as a focused analysis.

Minor concerns:

  1. Lines 528 and 529 on page 12 mixed the variables names and definitions of two variables.
  2. Please include the definitions of variables financial constraints and financialization in Table 1. If you consider the location of this table not being proper to include these two variables, then you can move it to an appendix.
  3. Make it clear what fixed effects are included in all regressions models. In some tables, no information is given. In some tables, year fixed effects are indicated. However, line 578 on page 13 mentions industry fixed effects. I believe that controlling differences across industries is very important with industry fixed effects, also the variable economic policy uncertainty must be highly correlated with year fixed effects.

Author Response

Dear Reviewer:

Thank you for your letter and for the reviewers’ comments concerning our manuscript sustainability-1575319 titled “Research on the Impact of Economic Policy Uncertainty on Enterprises' Green Innovation -Based on the perspective of corporate investment and financing decisions”. Those comments are all valuable and very helpful for revising and improving our paper, as well as the important guiding significance to our researches. We have studied comments carefully and have made correction which we hope meet with approval. Revised portion are marked in red in the revised manuscript.

The main corrections in the paper and the responds to the comments of reviewer are as flowing:

(1) The set-up of the manuscript is not engaging yet.

Response: The reviewer gave us a very professional and good suggestion. As reviewer’s suggestion, we rewrote the abstract and introduction in the revised article.

(2) The claims for the mediating roles of financial constraints and financialization in section 6, which is about the analysis on the transmission mechanism of enterprise green innovation affected by economic policy uncertainty, have not been grounded well. My suggestion to the authors is to convert the mediating role of financial constraints and financialization to moderating roles, which propose a joint effect of these two factors along with economic policy uncertainty on green innovation. This will generate a more reliable approach to follow the arguments as well as a focused analysis.

Response: We are very sorry that we did not make this clear in the previous article. In the revised article, according to the opinions of reviewer, we re-completed the section 6 by taking financing constraints and financialization as moderating variables of the impact of economic policy uncertainty on enterprise green innovation.

(3) Lines 528 and 529 on page 12 mixed the variables names and definitions of two variables.

Response: We are very sorry for our incorrect writing of the manuscript. We have modified the part in the revised article.

(4) Please include the definitions of variables financial constraints and financialization in Table 1. If you consider the location of this table not being proper to include these two variables, then you can move it to an appendix.

Response: Considering the reviewer’s suggestion, we have added the definition and calculation method of variables financing constraints and financialization in Table 1. And adjust Table 1 to the appendix at the end of the paper in the revised article.

(5)Make it clear what fixed effects are included in all regressions models.

Response: We are very sorry that we did not make this clear in the previous article. In this paper, the regression model contains annual fixed effects. We added fixed-effect information to every table. We apologize for a clerical error on page 13, line 578, which misstated annual fixed effect as industry fixed effect. We have made modifications in the revised draft.

We tried our best to improve the manuscript and made some changes in the manuscript. These changes will not influence the content and framework of the paper. And here we did not list all changes, but marked in red in revised paper.

We appreciate for Editors and Reviewers’ warm work earnestly, and hope that the correction will meet with approval.

Once again, thank you very much for your comments and suggestions.

Yours sincerely,

Wenjun Zhou

Reviewer 2 Report

  • At the end of the Introduction section I recommend you add a paragraph describing the structure of the paper
  • page 18, figure 4 it is a little out of place
  • please add in the conclusions section a paragraph describing the limits of the research you have done in this paper but also to point out some future research directions on this topic.
  • pay attention to how you format the reference list, to be a unitary format and according to the instructions provided by the journal.
  • Also, the list of references is a bit old, there are a limited number of studies from the last three or four years, so I recommend quoting more recent studies.

Author Response

Point 1: At the end of the Introduction section I recommend you add a paragraph describing the structure of the paper.

Response 1: The reviewer gave us a very professional and good suggestion. As reviewer’s suggestion, we rewrote the abstract in the revised article. The research framework, main research conclusions, research contributions and other relevant contents are added.

Point 2: Page 18, figure 4 it is a little out of place.

Response 2: We are very sorry that we did not make this clear in the previous article. We have redrawn the diagrams included in the article.

Point 3: Please add in the conclusions section a paragraph describing the limits of the research you have done in this paper but also to point out some future research directions on this topic.

Response 3: Considering the reviewer’s suggestion, we have added this part in the revised article. The limitations of the new study are as follows:

The research content of this article still has some limitations. In the future, the authors intend to further explore the causes and mechanisms of the impact of economic policy uncertainty on green innovation. This paper finds that the impact of economic policy uncertainty on green innovation presents an inverted U shape. However, there is no further analysis of the reasons for maintaining the inverted U-shaped relationship between them. Macroeconomic policy changes will affect enterprises' choice of green innovation behavior from many aspects. However, it is very difficult to find instruments affecting one of many endogenous variables, but not the other. This also makes it difficult to analyze the reasons and ways that economic policy uncertainty affects enterprise green innovation. Of course, this is also the direction and breakthrough point of the author's future research.

Point 4: Pay attention to how you format the reference list, to be a unitary format and according to the instructions provided by the journal.

Response 4: We apologize for the formatting error in the original manuscript. We re-examined the format of the references in the article and modified it according to the journal's requirements in the revised article.

Point 5: Also, the list of references is a bit old, there are a limited number of studies from the last three or four years, so I recommend quoting more recent studies.

Response 5: As suggested by the reviewer, we removed some older references. Some recently published research results are added and updated in reference articles.

Round 2

Reviewer 1 Report

I appreciate that the authors have attempted to respond to each of my comments carefully. 

Author Response

Point 1: In the tables of results, why in the brackets is there a negative sign before all the t-statistics?

Response 1: We are very sorry for our incorrect writing of the manuscript. The data in the tables was manually entered by the author in word document. A negative sign was mistakenly added in front of many t-statistics during the entry process. We rechecked the data in each table and made corrections. Revise the symbol before each t-statistics in Table 2- Table 5. Moreover, we also found that we erroneously wrote the values in brackets in Table 5 as standard errors. We make corrections in the new revision. The numerical errors in brackets in the tables do not affect the conclusion and the rest of the paper. We apologize for the carelessness in writing the paper again.

Point 2: Table 5 needs to be made more concise and smaller.
  Response 2: Considering the reviewer’s suggestion, we redesigned the format of Table 5 in the revised article. We simplified the names of the four sample groups and designed the header of Table 5 to be simpler.

Point 3: Although it reads well, the paper could do with a final proof reading before publication.

Response 3: Considering the reviewer’s suggestion, we recalibrated the paper for format, grammar and content.